# Association between Preoperative 18-FDG PET-CT SUVmax and Next-Generation Sequencing Results in Postoperative Ovarian Malignant Tissue in Patients with Advanced Ovarian Cancer

**DOI:** 10.3390/jcm12062287

**Published:** 2023-03-15

**Authors:** Jung Min Ryu, Yoon Young Jeong, Sun-Jae Lee, Byung Wook Choi, Youn Seok Choi

**Affiliations:** 1Department of Obstetrics and Gynecology, School of Medicine, Daegu Catholic University, Daegu 42472, Repulic of Korea; 2Department of Pathology, School of Medicine, Daegu Catholic University, Daegu 42472, Repulic of Korea; 3Department of Nuclear Medicine, School of Medicine, Daegu Catholic University, Daegu 42472, Repulic of Korea

**Keywords:** single nucleotide variants, insertions and deletions, next-generation sequencing, ovarian cancer, SUVmax, *TP53*

## Abstract

This study investigated the association between maximum standardized uptake values (SUVmax) on preoperative 18-FDG PET-CT and next-generation sequencing (NGS) results in post-surgical ovarian malignant tissue in patients with advanced ovarian cancer. Twenty-five patients with stage IIIC or IV ovarian cancer who underwent both preoperative 18-FDG PET-CT and postoperative NGS for ovarian malignancies were retrospectively enrolled. Two patients had no detected variants, 21 of the 23 patients with any somatic variant had at least one single nucleotide variant (SNV) or insertion/deletion (indel), 10 patients showed copy number variation (CNV), and two patients had a fusion variant. SUVmax differed according to the presence of SNVs/indels, with an SUVmax of 13.06 for patients with ≥ 1 SNV/indel and 6.28 for patients without (*p* = 0.003). Seventeen of 20 patients with Tier 2 variants had *TP53* variants, and there was a statistically significant association between SUVmax and the presence of *TP53* variants (13.21 vs. 9.35, *p* = 0.041). Analysis of the correlation between the sum of the Tier 1 and Tier 2 numbers and SUVmax showed a statistically significant correlation (*p* = 0.002; Pearson’s r = 0.588). In conclusion, patients with advanced ovarian cancer with SNVs/indels on NGS, especially those with *TP53* Tier 2 variants, showed a proportional association with tumor SUVmax on preoperative PET-CT.

## 1. Introduction

Patients with advanced-stage epithelial ovarian cancer show a poor prognosis [1]. The response to adjuvant chemotherapy after thorough optimal surgical debulking is an important determinant of prognosis [2]. In addition, with the recent development of molecular genetics, the presence and types of genetic mutations in patient genes and malignant tumors now comprise an important therapeutic field in cancer treatment [3]. Therefore, the importance of next-generation sequencing (NGS) results has recently been highlighted in the treatment of advanced ovarian cancer [4].

It is now possible to perform NGS rapidly and efficiently compared to direct sequencing. With NGS, both detailed diagnosis and customized treatment selection (based on individual cancer genome information) have become possible [3]. In gynecologic oncology, NGS is useful for detecting various types of mutations as well as for clinical management [5]. Genomic single nucleotide variants (SNV), insertions/deletions (indels), copy number variants (CNV), and fusion variants in somatic malignant tissues can be identified through NGS analysis [6].

By intravenously injecting a radiopharmaceutical that emits positrons into the body, ^18^F-fluoro-2-deoxy-d-glucose positron emission tomography-computed tomography (18-FDG PET/CT) imaging scans can display the distribution of the drug in a three-dimensional image [7]. This method is mainly used for the diagnosis of various malignancies and is useful for the differential diagnosis of cancer, staging, recurrence evaluation, and determination of treatment effectiveness along with tumor markers [8]. The maximum standardized uptake value (SUVmax) is a quantitative standard index representing the tissue glucose metabolism rate in PET/CT findings; it is a key value in the effective discrimination of malignancy [9].

The purpose of this study was to investigate the association between SUVmax as found in preoperative 18-FDG PET-CT and NGS results obtained from post-surgical malignant ovarian tissue in patients with advanced ovarian cancer.

## 2. Materials and Methods

From January 2019 to June 2022, patients with ovarian cancer who underwent both preoperative 18-FDG PET-CT and postoperative NGS of malignant ovarian tissue samples collected at Daegu Catholic University Hospital were enrolled in the present study. All enrolled patients consented to ovarian tissue collection and NGS. Patients who were enrolled met the following criteria: the ovary was the primary origin of malignancy, they had an epithelial-type cancer (high-grade serous carcinoma; mucinous, endometrioid, or clear cell carcinomas; or carcinosarcoma), they had International Federation of Gynecology and Obstetrics (FIGO) stage IIIC or IV disease, they underwent complete optimal debulking and staging surgery before NGS, and they received an adjuvant chemotherapy regimen based on paclitaxel and carboplatin. Each patient’s clinical characteristics, including age, final histopathologic biopsy results, the presence of pelvic and/or para-aortic lymph node metastasis, recurrence and death status, and NGS results were reviewed retrospectively using medical records. All histopathological and NGS results from malignant ovarian tumor specimens were reviewed by an expert gynecologic pathologist at Daegu Catholic University Hospital.

Areas of cancerous tissue in formalin-fixed, paraffin-embedded (FFPE) sections were identified by a specialized pathologist (S.-J. Lee); 10 unstained sections with a thickness of 10 μm per subject were prepared for the assay. The NGS process, from the extraction of DNA and RNA to the data analysis, was entrusted to a commercial laboratory (GC Labs, Yongin, Republic of Korea). Total DNA and RNA were extracted for NGS analysis using RecoverAll™ Total Nucleic Acid Isolation Kit for FFPE (Invitrogen™, Invitrogen, Life Technologies, Carlsbad, CA, USA) according to the manufacturer’s instructions. DNA and copy DNA (cDNA) libraries were prepared using the quality-controlled Oncomine Comprehensive Plus Ampilion-based assay panel (Thermo Fisher Scientific, Waltham, MA, USA), which includes a total of 425 tumor-associated genes, according to the manufacturer’s standard protocol.

Sequencing analysis provides information about somatic variants, including SNVs, indels, CNVs, and gene fusions. In this study, sequencing was performed by NGS using an Ion Torrent S5 System Sequencer (Thermo Fisher Scientific, Waltham, MA, USA); sequencing data was analyzed on a Torrent Server using Torrent Suite Software. Alignment to the hg19 human reference genome and further management was performed using Ion Reporter™ Software with the “Oncomine Comprehensive Plus—w2.3—DNA and Fusions—Single Sample” workflow. The criteria for variant calls were set as follows: SNV/indel, variant allele frequency of ≥5% in SNVs and ≥5% in indels; CNV, average CNVs of ≥4 (gain) and <1 (loss); translocations, read counts of ≥20; and ≥50,000 total valid mapped reads. CNVs were analyzed in samples with a median absolute value of all pairwise differences (MAPD) of <0.5, which represents a measure of read coverage noise detected across all amplicons in a sample. Variant characteristics were evaluated using Catalogue of Somatic Mutations in Cancer (COSMIC), cBioPortal, Cancer Hotspots, OncoKB, My Cancer Genome, Clinical Knowledgebase Browser databases, and National Comprehensive Cancer Network (NCCN) guidelines, and were classified by clinical significance using the abovementioned tier system [6].

^18^F-FDG PET/CT scans were performed using the Discovery IQ PET/CT scanner (GE Healthcare, Milwaukee, WI, USA). All patients fasted for at least 6 h before ^18^F-FDG injection, and each patient’s blood glucose concentration was confirmed to be < 150 mg/dL. Patients were administered 4.0 MBq/kg of ^18^F-FDG intravenously. Integrated PET/CT images were acquired from the base of the skull to the proximal thigh approximately 60 min after ^18^F-FDG administration. For PET/CT images, non-contrast-enhanced whole-body CT was performed using the following protocols: 120 kVp, smart auto mA (60–80 mAs adjusted to the patients’ body weight), 0.5-s rotation time, 3.75-mm helical thickness, 0.938 pitch, 18.75 mm/rotation table speed, and a 512 × 512 matrix. PET scans from the cerebellum to the proximal thigh were acquired after the CT scan, and the matrix size was 256 × 256. The acquisition time was 3 min per bed position. The PET images were reconstructed using the vendor-supplied Q. Clear technique, a block sequential regularized expectation maximization penalized-likelihood reconstruction algorithm (*β* = 350).

All ^18^F-FDG PET/CT images were retrospectively reviewed using a dedicated vendor-supplied workstation (GE Advantage Workstation version 4.7; GE Healthcare, Chicago, IL, USA). Two experienced nuclear medicine physicians interpreted the ^18^F-FDG PET/CT images of all patients without knowledge of their clinicopathologic information. An ellipsoidal volume of interest was manually drawn to encompass the primary ovarian cancer lesion on the fused PET/CT images, while excluding the physiologic uptake of adjacent organs and vessels by considering the tumor location. SUVmax was defined as the highest SUV of the primary tumor and was obtained using the following formula: SUVmax = maximum activity in the region of interest (MBq/g)/(injected dose [MBq]/body weight [g]).

Data were analyzed using IBM SPSS statistical software (V25.0; IBM, Armonk, NY, USA). Comparisons of variables between groups were based on independent samples t-tests and Pearson’s correlation tests. *p*-values were obtained using two-tailed tests; *p* < 0.05 was considered the threshold for statistical significance. As this study consisted of retrospective data collection, informed consent was not required. This retrospective study was approved by the Institutional Ethics Committee of Daegu Catholic University Hospital (approval number: CR-22-164) and was conducted in accordance with the principles of the Declaration of Helsinki.

## 3. Results

A total of 25 patients with stage IIIC or IV ovarian cancer were enrolled in this study. All enrolled patients underwent both preoperative 18-FDG PET-CT and postoperative NGS for ovarian malignancy at Daegu Catholic University Hospital (Daegu, Republic of Korea). The medical and demographic characteristics of the enrolled patients are summarized in Table 1. Of the 25 patients with ovarian malignancies, 18 had stage IIIC and seven had stage IV ovarian malignancies. Ten patients had pelvic lymph node metastasis and 12 patients had para-aortic lymph node metastasis. The final histopathological findings of patients with ovarian malignancy were as follows: high-grade serous carcinoma (*n* = 17), mucinous cystadenocarcinoma (*n* = 1), endometrioid carcinoma (*n* = 1), clear cell carcinoma (*n* = 2), and carcinosarcoma (*n* = 4).

NGS results, including for SNVs/indels, CNVs, and fusion variants, were reported in this research. Of the 25 enrolled patients, two patients had no variants and 23 patients had at least one SNV/indel, CNV, and/or fusion variant (Table 2). Of the 23 patients with variants, 21 had at least one SNV/indel variant, 10 had a CNV variant, and two had a fusion variant. Based on guidelines published in 2017, the SNV/indel variant results were classified into a tier system [6]. Specifically, the details of Tiers 1–4 are as follows: Tier 1 represents FDA-approved variants of strong clinical significance in therapeutic, prognostic, and diagnostic settings; Tier 2 represents variants of potential clinical significance; Tier 3 represents variants of unknown clinical significance; and Tier 4 represents variants that are either benign or likely benign. Variants classified as Tier 4 were excluded from the present study.

The average SUVmax values for each variant on 18-FDG PET-CT are compared in Table 2; these means were analyzed using independent t-tests. According to the tier classification system of SNVs/indels, there were eight patients with Tier 1 variants, 20 patients with Tier 2 variants, three patients with Tier 3 variants, and no patients with Tier 4 variants. A total of 21 patients had either Tier 1 or 2 variants. The SUVmax showed a statistically significant difference (*p* = 0.003) between those with or without SNVs/indels, with an SUVmax of 13.06 for patients with at least one SNV/indel variant and 6.28 for patients without an SNV/indel variant (Figure 1). The SUVmax of patients with Tier 2 variants was 13.21, but patients without Tier 2 variants had an SUVmax of 7.45, showing a statistically significant difference (*p* = 0.008). In addition, 17 out of 20 patients with Tier 2 variants had *TP53* variants, and there was a statistically significant association between SUVmax and the presence of *TP53* variants (13.21 vs. 9.35, *p* = 0.041).

As shown in Figure 2, a Pearson’s correlation test was performed to evaluate the correlation between the sum of Tier 1 and Tier 2 numbers and SUVmax; the correlation was statistically significant (*p* = 0.002). The Pearson’s correlation coefficient was 0.588, indicating a moderate correlation. However, there was no statistically significant difference in SUVmax between single tiers of SNV/indel, CNV, or fusion variants.

Table 3 analyzes the detailed variants among the identified Tier 1 or 2 variants (SNVs/indels). The Tier 1 SNV/indel variants were *KRAS*, *BRCA1*, and *BRCA2*. Among the Tier 1 and 2 variants, the most common variant was *TP53*. The detected Tier 1 variants were as follows: two *KRAS* variants (8%) and three *BRCA1* or *BRCA2* variants (12%). Tier 2 variants were as follows: 17 *TP53* variants (68%), two *PIK3CA* variants (8%), and one (4%) variant each for *PTEN*, *FBXW7*, *ESR1*, *FGFR2*, *SPOP*, *NOTCH3*, *MSH3*, *FGFR4*, *NF1*, *CREBBP*, and *ARID1A*.

## 4. Discussion

In our study, we found a correlation between the presence of SNV/indel mutations in patients with advanced ovarian cancer, particularly those with Tier 2 *TP53* variants, and the SUVmax of tumors on preoperative PET/CT. Given recent developments in molecular genetics, immunological examinations, and cancer treatments, both poly (ADP-ribose) polymerase (PARP) inhibitors and immune checkpoint inhibitors are being actively studied as treatment options that could be used alongside chemotherapy or radiation therapy in gynecological oncology [5,10]. Among the various available methods, malignant tissue-based NGS detects a number of genomic mutations and has clinical applications based on its specific results. Currently, both germline *BRCA* analysis (via blood tests) and NGS results (from malignant tissue obtained post-surgery) are tests used to indicate PARP inhibitor treatment. Both disease-free survival and overall survival are improved when a PARP inhibitor is applied to patients with germline or somatic *BRCA* mutations [11,12]. Similarly, the clinical application of immune checkpoint inhibitors is based on either the microsatellite instability/mismatch repair (MSI/MMR) test or the degree of programmed death-ligand 1 (PD-L1) expression in malignant tissues [13,14]. However, other than the instances mentioned above, to date applications of PARP inhibitors or immune checkpoint inhibitors are not well elucidated in patients with advanced ovarian cancer.

Although PET/CT has limitations in detecting small metastatic lesions or carcinomatosis, PET/CT is frequently used to both preoperatively predict clinical stage in patients with suspected ovarian cancer and determine the extent of surgery [9]. Among the values obtained by converting the degree of FDG uptake into a quantitative parameter at the primary tumor site through PET/CT, SUVmax is the most frequently used value. This value is used as a representative value among various parameters that predict malignancy preoperatively. In a previous study, high SUVmax levels in epithelial ovarian cancer were associated with chemosensitivity and proliferation [15]. In addition, other studies have demonstrated that SUVmax is useful for evaluating radiosensitivity [16]. However, there are several conflicting findings regarding the association between SUVmax and prognosis. Some studies have shown that a high SUVmax is associated with chemosensitivity and radiosensitivity and results in a good prognosis, while other studies have shown that tumors with a high SUVmax are poorly differentiated and have a poor prognosis [17,18,19]. Therefore, the association between SUVmax and prognosis remains unclear.

NGS detects alterations, deletions, duplications, insertions, inversions, rearrangements, translocations, or combinations of these genetic changes in genomic segments. NGS results include readings for SNV/indel, CNV, and fusion variants. We note that the most valuable findings are usually determined from SNV/indel results [20]. SNVs and indels are known to cause various diseases and malignancies. Therefore, determining accurate SNVs and indels from NGS test results is a key analysis performed in genome research and is a long-term area of study for research on various diseases. SNVs can occur when a single nucleotide (A, T, C, or G) is altered in a DNA genomic sequence, whereas an indel refers to a small length of DNA (usually < 50 base pairs) inserted or deleted in the genome. As mentioned above, SNV/indel results are classified into variants from Tier 1 to Tier 4 according to their clinical significance; those from Tier 1 and Tier 2 variants are generally clinically important.

In our study, patients with advanced ovarian cancer with Tier 1 and/or Tier 2 SNVs (according to NGS) were proportional to the SUVmax of the tumor on preoperative PET-CT. Therefore, it might be inferred that advanced ovarian cancer patients with Tier 1 and/or 2 SNVs are likely to be affected in the chemosensitivity, radiosensitivity, and tumor proliferation domains [15,16]. A study found a statistically significant association between 18F-FDG uptake and PD-L1 expression in malignant tissues; this suggests that PET/CT may play an important role in the selection of ovarian cancer patients for anti-PD-L1 antibody therapy [21]. In practice, however, to our knowledge there has been no previous study investigating whether patients with high SUVmax on PET-CT respond better to PD-1 blocker immune checkpoint inhibitors, such as pembrolizumab. In addition, in previous studies SNVs have also been associated with microsatellite instability depending on variant location, suggesting that their presence may result in the accumulation of gene mutations that generate neoantigens that may be more responsive to immune checkpoint inhibitors [22,23]. To date, only the MSI/MMR test and PD-L1 immunohistochemistry have been used as indicators for pembrolizumab. Therefore, SNVs are associated with MSI, and it may be presumed that patients with high SUVmax values in preoperative PET-CT tumors are likely to have a better response to immune checkpoint inhibitors such as pembrolizumab, because they are associated with clinically important Tier 1 and 2 genomic SNVs.

In one study, malignant tumors derived from breast, lung, and colorectal cancer patients with *TP53* mutations showed higher SUVmax values than those from patients without *TP53* mutations [24]. In addition, another study found an association between the total number of genomic alterations and SUVmax values in malignant tissue biopsies from patients with breast, gastrointestinal, and lung cancers [25]. However, these studies were limited in that the default SUVmax value differed for each primary tumor site. Compared to that study, the results of our study are valuable in that we only investigated the advanced stages of ovarian cancer, specifically epithelial ovarian cancer. In another study on breast cancer, there was an association between *BRCA* variants detected via NGS and SUVmax values [26]. However, to our knowledge, no studies have found an association between NGS results and SUVmax values in ovarian cancer, as in our study.

Several previous studies have investigated the correlation between SUVmax and gene variants [25,27]. According to these studies, SUVmax is positively associated with the number of total oncogenic variants. Mutations in a high number of oncogenes lead to metabolic reprogramming by both stimulating glucose uptake and channeling glucose through aerobic glycolysis [28]. The correlation is also involved in the pentose phosphate pathway, gluconeogenesis and glycogen metabolism, and is associated with increased SUVmax. In another study, a large number of genomic variants involved in elevated circulating tumor DNA first promoted metabolic reorganization, then increased glucose metabolism, and finally increased SUVmax values [29]. In addition, studies have also shown that increased genetic mutation rates lead to a more aggressive tumor phenotype and are associated with an increase in FDG uptake [30,31,32].

Some limitations of this study are that it was a retrospective study conducted at one hospital center, and it included only a small number of patients. However, as the first study on the association between SUVmax of preoperative 18 FDG PET-CT and postoperative NGS in patients with advanced epithelial ovarian cancer, it has clinical value. In addition, few studies have investigated NGS finings, SUVmax detected via PET/CT, and response to immune checkpoint inhibitors. Based on the results of the present study, there is an association between preoperative PET/CT SUVmax and the presence of SNVs in postoperative NGS findings, and it is possible that these findings have implications with respect to a better response to immune checkpoint inhibitors. However, further research is needed.

## 5. Conclusions

In conclusion, patients with advanced ovarian cancer with SNVs/indels on NGS, especially those with *TP53* Tier 2 variants, showed a proportional association with tumor SUVmax on preoperative PET-CT.

## Figures and Tables

**Figure 1 jcm-12-02287-f001:**
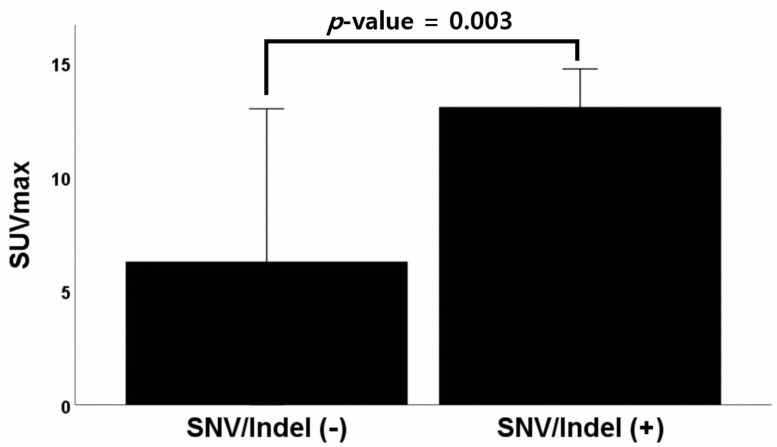
Graph of the difference in average SUVmax between patients with and without SNV/Indel variants. Abbreviations: SUVmax, maximum standardized uptake value; SNV, single nucleotide variants; Indel, insertions and deletions.

**Figure 2 jcm-12-02287-f002:**
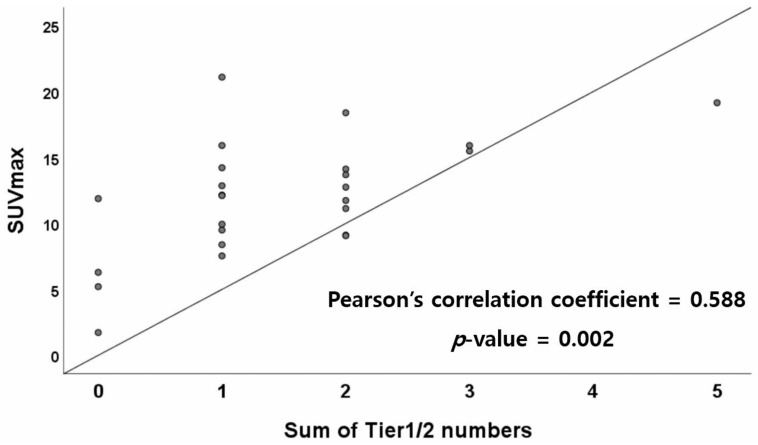
Correlation between the sum of Tier 1 and Tier 2 numbers and SUVmax. Abbreviations: SUVmax, maximum standardized uptake value.

**Table 1 jcm-12-02287-t001:** Histopathology and characteristics of enrolled patients with ovarian malignancy (*n* = 25).

Characteristics	Number of Patients
Age	
<50	10 (40%)
≥50	15 (60%)
Stage	
IIIC	18 (72%)
IV	7 (28%)
Pelvic LN metastasis	
Yes	10 (40%)
No	15 (60%)
Para-aortic LN metastasis	
Yes	12 (48%)
No	13 (52%)
Recurrence	
Yes	15 (60%)
No	10 (40%)
Death	
Yes	4 (16%)
No	21 (84%)
Histopathology	
HGSC	17 (68%)
Mucinous	1 (4%)
Endometrioid	1 (4%)
Clear cell	2 (8%)
Carcinosarcoma	4 (16%)

Abbreviations: LN, Lymph node; HGSC, High grade serous carcinoma.

**Table 2 jcm-12-02287-t002:** Mean and standard deviation of SUVmax according to each variant type found in NGS results.

NGS Results (Total 25)	SUVmax (Mean ± SD)	*p*-Value
Variant not found (*n* = 2)	8.55 ± 4.72	*p* = 0.265
Variant found (*n* = 23)	12.27 ± 4.41	
SNV/Indel not found (*n* = 4)	6.28 ± 4.21	*p* = 0.003
SNV/Indel found (*n* = 21)	13.06 ± 3.66	
Tier1 not found (*n* = 17)	11.82 ± 5.08	*p* = 0.804
Tier1 found (*n* = 8)	12.31 ± 3.00	
BRCA 1/2 not found (*n* = 19)	11.69 ± 4.83	*p =* 0.582
BRCA 1/2 found (*n* = 6)	12.87 ± 3.18	
Tier2 not found (*n* = 5)	7.45 ± 4.49	*p* = 0.008
Tier2 found (*n* = 20)	13.11 ± 3.75	
*TP53* not found (N = 8)	9.35 ± 4.35	*p* = 0.041
*TP53* found (*n* = 17)	13.21 ± 4.06	
Tier3 not found (N = 22)	11.56 ± 4.46	*p* = 0.220
Tier3 found (*n* = 3)	14.98 ± 3.69	
CNV not found (*n* = 15)	11.77 ± 3.72	*p* = 0.789
CNV found (*n* = 10)	12.28 ± 5.59	
Fusion not found (*n* = 23)	11.79 ± 4.35	*p* = 0.487
Fusion found (*n* = 2)	14.13 ± 7.10	

Abbreviations: SUVmax, maximum standardized uptake value; NGS, next-generation sequencing; SD, standard deviation; SNV, single nucleotide variants; Indel, insertions and deletions; CNV, copy number variation.

**Table 3 jcm-12-02287-t003:** Frequency of SNV/Indel variants in Tiers 1 and 2 found in NGS results.

Gene Variants	*n*
Tier 1	
KRAS	2 (8%)
BRCA1	3 (12%)
BRCA2	3 (12%)
Tier 2	
TP53	17 (68%)
PIK3CA	2 (8%)
PTEN	1 (4%)
FBXW7	1 (4%)
ESR1	1 (4%)
FGFR2	1 (4%)
SPOP	1 (4%)
NOTCH3	1 (4%)
MSH3	1 (4%)
FGFR4	1 (4%)
NF1	1 (4%)
CREBBP	1 (4%)
ARID1A	1 (4%)

## Data Availability

Data for this study, though not available in a public repository, will be made available to other researchers upon reasonable request.

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
