# Peer review of "Association between Preoperative 18-FDG PET-CT SUVmax and Next-Generation Sequencing Results in Postoperative Ovarian Malignant Tissue in Patients with Advanced Ovarian Cancer"

_jcm, 2023, doi:10.3390/jcm12062287_

Round 1

Reviewer 1 Report

The paper is very interesting and thank you for the opportunity to review it.

I would like to know how you decide to make this comparison between  Preoperative 18-FDG PET-CT SUVmax and next-generation Sequencing Results?

Do you think tumoral markers could be useful in addition to PET-CT? If yes take into account these papers: doi: 10.3390/jcm1107199; doi: 10.3389/fsurg.2022.1068492.

Why did you decide to include only patients stage IIIC and not IIIA or B?

Regarding the first part of discussion on PARP-I I suggest it (DOI: 10.1080/13543784.2021.1901882)

I suggest to include a subparagraph on strenghts and limitation of our study

Author Response

<Reviewer 1>

The paper is very interesting and thank you for the opportunity to review it.

I would like to know how you decide to make this comparison between  Preoperative 18-FDG PET-CT SUVmax and next-generation Sequencing Results?

  • First of all, I am really appreciating you for reviewing our research paper. This paper is about the association between SUVmax of 18 FDG PET-CT before surgery and NGS (next-generation sequencing) after surgery in patients with advanced epithelial ovarian cancer. It has not been long since the NGS was introduced in the ovarian cancer, and since both PET-CT and NGS are expensive test, so this subject has not been well addressed in other studies. Therefore, we judged that this topic might have clinical value.

Do you think tumoral markers could be useful in addition to PET-CT? If yes take into account these papers: doi: 10.3390/jcm1107199; doi: 10.3389/fsurg.2022.1068492.

  • You suggested two references. However, the first reference (doi: 10.3390/jcm1107199) does not appear even after searching. I wonder if there is a typo in the numbers.
  • In the case of the second suggested reference, we have added a reference to the manuscript, stating that it is more helpful for diagnosis to evaluate a combination of multiple tumor markers than a single tumor marker.

Why did you decide to include only patients stage IIIC and not IIIA or B?

  • In the case of ovarian cancer, it is often diagnosed as stage IIIC at the first diagnosis. In our center, the numbers of stage IIIA and IIIB were very limited. In addition, since the majority of patients diagnosed with ovarian cancer as stage IIIC after staging operation, only stage IIIC patients were included.

Regarding the first part of discussion on PARP-I I suggest it (DOI: 10.1080/13543784.2021.1901882)

  • The reference you suggested was judged to be appropriate. I decided that it would be good to add it to the first paragraph of discussion, so I added it.

I suggest to include a subparagraph on strenghts and limitation of our study

  • This study is a retrospective study conducted at a unit center, and the number of patients included in the study is limited is considered a limitation of this study. However, as this is the only paper researched on this topic, so we judged that it has novelty. This point is emphasized in the last paragraph of the discussion. Thank you for the good review.

Reviewer 2 Report

I read with interest this article. It is well designed and well reported. However , what is the main point of this study ?

What this study adds ? Can this information be added as guidelines ?

Author Response

<Reviewer 2>

I read with interest this article. It is well designed and well reported. However , what is the main point of this study ?

What this study adds ? Can this information be added as guidelines ?

  •  First of all, I am really appreciating you for reviewing our paper. This paper is about the association between SUVmax of 18 FDG PET-CT before surgery and NGS (next-generation sequencing) after surgery in patients with advanced epithelial ovarian cancer. This study is a retrospective study conducted at one unit center, and it is difficult to change the current guideline with this result alone. However, the correlation between preoperative PET-CT of ovarian cancer patients and NGS results in tissue obtained after surgery is a novel topic that has not been studied before. Although additional research is definitely needed, based on these results, in addition to the results known so far such as MSI, it also suggests that there might that it may be related to the response of immune check point inhibitors according to PET-CT or NGS results in the future. Also, It has not been long since the NGS was introduced in the ovarian cancer, and since both PET-CT and NGS are expensive test, so this subject has not been well addressed in other studies. Therefore, authors judged that this topic might have clinical value, and submitted the paper.

Thank you for your review. Please let me know if there are any additional corrections.

Sincerely,

Jung Min Ryu